# CKII Control of Axonal Plasticity Is Mediated by Mitochondrial Ca^2+^ via Mitochondrial NCLX

**DOI:** 10.3390/cells11243990

**Published:** 2022-12-09

**Authors:** Tomer Katoshevski, Lior Bar, Eliav Tikochinsky, Shimon Harel, Tsipi Ben-Kasus Nissim, Ivan Bogeski, Michal Hershfinkel, Bernard Attali, Israel Sekler

**Affiliations:** 1Department of Physiology and Cell Biology, Ben Gurion University, Beer-Sheva 8410501, Israel; 2Department of Physiology & Pharmacology, Sackler Faculty of Medicine, Tel-Aviv University, Tel-Aviv 6997801, Israel; 3Department of Cardiovascular Physiology, Göttingen University, 37073 Göttingen, Germany

**Keywords:** NCLX, mitochondrial Ca^2+^ signaling, CKII, neuronal plasticity

## Abstract

Mitochondrial Ca^2+^ efflux by NCLX is a critical rate-limiting step in mitochondria signaling. We previously showed that NCLX is phosphorylated at a putative Casein Kinase 2 (CKII) site, the serine 271 (S271). Here, we asked if NCLX is regulated by CKII and interrogated the physiological implications of this control. We found that CKII inhibitors down-regulated NCLX-dependent Ca^2+^ transport activity in SH-SY5Y neuronal cells and primary hippocampal neurons. Furthermore, we show that the CKII phosphomimetic mutants on NCLX inhibited (S271A) and constitutively activated (S271D) NCLX transport, respectively, rendering it insensitive to CKII inhibition. These phosphomimetic NCLX mutations also control the allosteric regulation of NCLX by mitochondrial membrane potential (ΔΨm). Since the omnipresent CKII is necessary for modulating the plasticity of the axon initial segment (AIS), we interrogated, in hippocampal neurons, if NCLX is required for this process. Similarly to WT neurons, NCLX-KO neurons can exhibit homeostatic plasticity following M-channel block. However, while WT neurons utilize a CKII-sensitive distal relocation of AIS Na^+^ and Kv7 channels to decrease their intrinsic excitability, we did not observe such translocation in NCLX-KO neurons. Thus, our results indicate that NCLX is regulated by CKII and is a crucial link between CKII signaling and fast neuronal plasticity.

## 1. Introduction

From apoptosis suppression of cancer cells [1,2,3] to abnormal protein phosphorylation in Alzheimer’s disease [4,5], the ubiquitous CKII takes part in many crucial physiological and pathophysiological processes in the cell. Lately, several studies have suggested that CKII might also have a metabolic role, which is still poorly understood [6,7,8]. In our previous study on PKA regulation of the mitochondrial Na^+^/Ca^2+^ exchanger, NCLX, we conducted a mass spectral analysis [9] that identified a putative CKII phosphorylation site residing on the regulatory loop of NCLX. Mitochondrial Ca^2+^ ([Ca^2+^]m) signaling is largely regulated by NCLX [10,11]. Following its flow into the mitochondria through the uniporter, Ca^2+^ is then extruded by the much slower NCLX, which is, therefore, a rate-limiting step in [Ca^2+^]m transport [12]. [Ca^2+^]m signaling plays several key roles in mitochondrial and cellular function; among them is the regulation of the Krebs cycle [13,14,15] and the electron transport chain [16], as well as tuning of local Ca^2+^ signaling hot spots at the vicinity of the cell membrane or the ER [17,18,19,20]. Moreover, imbalance in [Ca^2+^]m, the so-called [Ca^2+^]m overload, is a hallmark event of many major health syndromes such as neurodegenerative diseases, brain stroke, and cardiac failure [21]. CKII is a ubiquitous kinase and is a heterodimer composed of α1, α2, β1 and β2 subunits. CKII activity promotes cell proliferation and survival, and is therefore often linked to cancer propagation. This kinase dynamically shuttles between the cytosol and the nucleus. The strong antiapoptotic role of CKII is also attributed to the regulation of mitochondrial cell death-controlling proteins, such as BID and BCL2. However, it is unknown if CKII can enter the mitochondria or phosphorylate intra-mitochondrial proteins [22]. While most studies on CKII are focused on cancer, new and surprising studies have suggested that CKII also takes part in axonal plasticity [23]. The axon initial segment (AIS), the spike generation region of the neuron, can exhibit fast plasticity over time [24], a vital process for dynamic neuronal network tuning. AIS plasticity can be triggered by blocking axonal M-type K^+^ channels, leading to distal relocation of Na^+^ and M-channels on the AIS away from the soma within a few hours [25]. Although AIS plasticity is CKII-dependent, the link between CKII and this homeostatic neuronal process is poorly understood.

In this study, we first examined the role of the putative phosphorylation of NCLX by CKII. We found that CKII inhibitors downregulated mitochondrial Ca^2+^ efflux of NCLX. Furthermore, we produced and tested two NCLX phosphomimetic mutants on the NCLX CKII phosphorylation site that mimicked constitutive phosphorylation or the de-phosphorylation state of NCLX and found that they constitutively activate or inactivate NCLX-dependent [Ca^2+^]m efflux, respectively. Lastly, we asked if CKII-dependent AIS plasticity is mediated by NCLX, and found that it is blocked in NCLX-KO hippocampal neurons. Thus, our results indicate that CKII is a key regulator of mitochondrial Ca^2+^ by controlling NCLX and, thereby, AIS plasticity and neuronal excitability.

## 2. Results

### 2.1. Inhibition of CKII by TBB and TBCA Modulates the [Ca^2+^]m Transient

The regulatory domain of NCLX has a putative CKII phosphorylation site at Ser271 (Figure 1A,B), identified through mass spectral analysis in previous work ([9], see Figure 3, Table 1). To test the role of CKII in NCLX regulation, we first asked if inhibition of CKII would modulate [Ca^2+^]m efflux through NCLX by comparing [Ca^2+^]m transport in control SH-SY5Y cells versus SH-SY5Y cells treated with the CKII inhibitors Tetrabromocinnamic acid (TBCA) [26] or 4,5,6,7-tetrabromobenzotriazole (TBB) [27]. SH-SY5Y cells, co-expressing a mitochondrial-targeted Ca^2+^ sensor Cepia2mt and NCLX, were treated with TBCA (10 μM) or TBB (10 μM), and Ca^2+^ response was triggered by the application of ATP (100 μM). Addition of ATP triggers, as expected [9,28,29], a [Ca^2+^]m transient. Ca^2+^ efflux rate which was reduced in TBCA treated cells (57 ± 9% of NCLX^WT^) followed by similar inhibition by TBB [Ca^2+^]m efflux (56 ± 10% of NCLX^WT^) (Figure 1C,E). Both inhibitors displayed a similar inhibitory effect on Ca^2+^ efflux; however, the [Ca^2+^]m influx amplitude was decreased in cells treated with TBCA (79 ± 4% of NCLX^WT^) (Figure 1D). The basal intensity was also affected in cells treated with TBB (41 ± 8% of NCLX^WT^) (Figure 1F). Due to this effect and because TBCA is more selective to CKII, the following experiments were conducted with TBCA. However, the results of this set of experiments indicate that inhibition of CKII attenuated [Ca^2+^]m efflux. To determine if CKII inhibition had an effect on global Ca^2+^ signaling, we compared the cytosolic Ca^2+^ response, via Fluo-4 dye (2 µM) and triggered by ATP (100 μM) in the presence or absence of the CKII inhibitor TBCA (10 µM). There was no significant difference in the cytosolic Ca^2+^ response in TBCA-treated cells versus the control cells (Appendix A).

We also tested the effect of CKII inhibitor TBCA on NCLX wild-type primary hippocampal neurons derived from mice. Hippocampal neurons from mice expressing the fluorescent [Ca^2+^]m sensor protein GCaMP6f [30] were treated with TBCA and stimulated at 2 Hz for 1.3 s to generate an action potential (AP). NCLX wild-type neurons treated with TBCA exhibited a much lower [Ca^2+^]m efflux rate (16 ± 16% of NCLX-WT UT) (Figure 1G,I) compared to untreated cells. Furthermore, the [Ca^2+^]m influx amplitude was also decreased in TBCA-treated neurons (51 ± 6% of NCLX-WT UT) (Figure 1G,H), similarly to the TBCA-treated SH-SY5Y cells. These results suggest that inhibition of CKII also has a direct effect on mitochondrial influx by MCU. Alternatively, the stronger effect on mitochondrial [Ca^2+^]m efflux may have triggered an initial overload of [Ca^2+^]m that downregulated MCU activity and eventually lowered the [Ca^2+^]m influx amplitude apex [31,32].

### 2.2. NCLX S271 Mutants Mimic a Phosphorylated State of the CKII Site

Regulation of [Ca^2+^]m efflux by CKII may be triggered by a direct or indirect effect of CKII on this exchanger. To determine if CKII is acting directly on NCLX, we used the phosphomimetic strategy that we have successfully used previously to study the regulation of NCLX by PKA [9,12]. We generated two point mutations on NCLX Ser271: replacing Ser with Asp, creating a site that mimicked a constitutively phosphorylated state, NCLX^S271D^; and replacing Ser with Ala, creating a mutant that mimicked a constitutive dephosphorylated state, NCLX^S271A^. NCLX mutants were co-expressed with the shNCLX (Sigma-Aldrich) construct, aimed to silence endogenous NCLX activity without affecting the expression of the mutants in SH-SY5Y cells [9]. To monitor the expression of the mutants versus NCLX^WT^, we performed Western blot analysis of the c-Myc-tagged constructs using a c-Myc antibody and found that the expression of NCLX mutants and NCLX^WT^ were not significantly different (For Antibodies see Table 1). No expression of tagged NCLX was monitored, as expected, in the control vector (pcDNA3.1+) transfected cells (Figure 2A,B). Note that the NCLX band migrated as a dimer of ~100 kDa consistent with our pervious findings [28]. Their [Ca^2+^]m transient activity was then tested using the same experimental paradigm described in Figure 1. The [Ca^2+^]m removal rate by cells expressing mutant NCLX^S271A^ was slower compared to that of the NCLX^WT^ expressing cells (51 ± 10% of NCLX^WT^), demonstrating an inhibited NCLX efflux (Figure 2C,E). On the other hand, cells expressing the NCLX^S271D^ mutant did not display an inhibitory effect of [Ca^2+^]m removal and manifested the same rate as that of WT NCLX (91% ± 11% of NCLX^WT^) (Figure 2C,E).

Consistent with cells treated with TBCA in Figure 1, [Ca^2+^]m influx amplitude of the NCLX^S271A^ mutant was also significantly lower compared to NCLX^WT^ (49 ± 6% of NCLX^WT^) (Figure 2C,D). This effect was not apparent in the NCLX^S271D^ mutant (99 ± 11% of NCLX^WT^). Moreover, we do not see any effect on [Ca^2+^]m basal intensity (Appendix A).

If CKII phosphorylates S271 to regulate NCLX activity, then we predicted that the [Ca^2+^]m efflux activity of the phosphomimetic mutants, in contrast to NCLX^WT^, will be insensitive to a CKII inhibitor. To test this hypothesis, SH-SY5Y cells co-expressing Cepia2mt and NCLX^WT^ or NCLX^S271A^\NCLX^S271D^ mutants were treated with TBCA, and the [Ca^2+^]m efflux was then monitored (Figure 2F–H). No significant difference was observed between the untreated NCLX^WT^ expressing cells and the transfected NCLX^S271D^ cells (Figure 2F,H), and treatment with TBCA did not downregulate the [Ca^2+^]m efflux. Furthermore, TBCA treatment did not modulate the reduced [Ca^2+^]m efflux activity in NCLX^S271A^ expressing cells (NCLX^S271A^ TBCA treated 60 ± 15% of NCLX^WT^, and un-treated 68 ± 10% of NCLX^WT^) (Figure 2G,H), which were similar to rates measured in NCLX^WT^ expressing cells treated with TBCA (55 ± 8% of NCLX^WT^). Altogether, the results of this section indicate that NCLX activity is regulated by CKII phosphorylation of Ser271 on the exchanger.

### 2.3. Role of NCLX CKII Regulatory Site in Controlling the Allosteric Regulation of NCLX by ΔΨm

As we have previously shown, NCLX is allosterically regulated by slight changes in ΔΨm [12], and phosphorylation of NCLX by PKA takes part in this regulation [9]. Here, we examined if this allosteric regulation of NCLX by ΔΨm is controlled by the CKII phosphorylation site. First, we tested if the expression of the NCLX mutations modulated ΔΨm measured in cells loaded and perfused with Tetramethylrhodamine Methyl Ester Perchlorate (TMRM). We calibrated ΔΨm resting values by using the mitochondrial oxidative phosphorylation uncoupler, protonophore 2-[2-[4-(trifluoromethoxy)phenyl] hydrazinylidene]-propanedinitrile (FCCP) [33]. No significant difference in basal ΔΨm was observed between NCLX^WT^ and NCLX^S271A^\NCLX^S271D^ expressing cells (Figure 3A,B). We then pretreated the SH-SY5Y cells, co-expressing a Cepia2mt and NCLX^WT^ or NCLX mutant NCLX^S271A^\ NCLX^S271D^, with N5,N6-bis(2-Fluorophenyl)[1,2,5]oxadiazolo[3,4-b]pyrazine-5,6-diamine (BAM15) for 15 min. BAM15 is a mild mitochondrial uncoupler that enables precise and small changes in ΔΨm compared to FCCP [34]. Following BAM15 pretreatment, the [Ca^2+^]m transient was monitored as described in Figure 1. Consistent with our previous study, we found that the NCLX-dependent Ca^2+^ efflux rate was attenuated in the partially depolarized NCLX^WT^ expressing cells (52 ± 12% of NCLX^WT^) (Figure 3C,F). We then asked if phosphomimetic NCLX mutants, NCLX^S271D^ and NCLX^S271A^, modulate NCLX sensitivity to changes in ΔΨm. No inhibition was observed following partial depolarization in cells expressing the NCLX^S271D^ constitutive active mutant, as untreated NCLX^S271D^ expressing cells showed similar efflux rates to those of the treated cells (Figure 3D,F). In contrast, the [Ca^2+^]m efflux in NCLX^S271A^ expressing cells was attenuated to a similar level in cells both treated with BAM15 (18 ± 11% of NCLX^WT^) and untreated cells (56 ± 8% of NCLX^WT^) (Figure 3E,F). Our results, therefore, indicate that the CKII site participates in tuning NCLX activity, which is linked to its allosteric regulation by ΔΨm.

### 2.4. Intrinsic Excitability Properties of Hippocampal Neurons Derived from WT and NCLX-KO Mice

Mitochondrial activity is crucial for regulating neuronal excitability [17,35,36,37,38,39], and was identified as a central regulator of firing rate set points in hippocampal circuits [40]. Therefore, we examined the intrinsic properties of cultured excitatory and inhibitory hippocampal neurons derived from NCLX-WT and NCLX-KO mice. Whole-cell patch-clamp recordings were performed on cultured hippocampal neurons 13–15 days in vitro (DIV) to study the threshold current of the 2 ms injected depolarizing current necessary to evoke a single spike (Figure 4A). To identify GABAergic inhibitory neurons, we infected hippocampal cultures with a recombinant virus derived from an AAV-viral vector driving the expression of the fluorescent protein mCherry under the control of the specific GABAergic hDlx promoters [23,41] (Figure 4A). Excitatory hippocampal neurons were negative to mCherry fluorescence. Our results show that the threshold currents were similar in excitatory and inhibitory neurons, but were significantly lower in neurons derived from NCLX-KO mice compared to those from NCLX-WT mice (Figure 4B,C). This result suggests that hippocampal neurons derived from NCLX-KO mice are more excitable than those from NCLX-WT mice.

### 2.5. Plasticity of the Axon Initial Segment in Hippocampal Neurons Derived from WT and NCLX-KO Mice

We previously showed that chronic neuronal hyperactivity, induced by M-channel inhibition, triggered intrinsic homeostatic plasticity in cultured hippocampal excitatory neurons on a fast timescale (1–4 h), including changes in the threshold current and a distal relocation of FGF14 along the axon initial segment (AIS) [23,25]. In contrast, homeostatic adaptations of intrinsic excitability failed in hippocampal GABAergic inhibitory neurons, which remained hyperexcitable following chronic M-channel blockage [23]. Here, we examined the impact of chronic M-channel inhibition on cultured hippocampal neurons derived from NCLX-KO mice. We found that acute M-channel block by 10 µM XE991 increased the excitability of both excitatory and inhibitory hippocampal neurons of NCLX-KO mice by significantly lowering the value of the threshold current (from 626 ± 37 pA to 459 ± 37 pA in excitatory neurons, and from 602 ± 35 pA to 416 ± 30 pA in inhibitory neurons) (Figure 4D). However, 4 h of chronic treatment with 10 µM XE991 triggered homeostatic adaptation of both excitatory and inhibitory neurons, with a return of the threshold current to the initial values measured in untreated hippocampal neurons (Figure 4D). Since we previously found that the AIS plasticity was dependent on CKII activity [25], we next examined the AIS location of FGF14 following 4 h of chronic XE991 treatment of excitatory and inhibitory hippocampal neurons, as well as the effect of blocking of CKII activity by tetrabromocinnamic acid (TBCA). FGF14 is a protein physically bridging Nav1.6 and Kv7.2 channels along the AIS, and therefore represents a valuable reporter of the location of Nav1.6 and Kv7.2 channels at the AIS [42]. We focused on pyramidal-like excitatory neurons, which represent the majority (≈80%) of cultured hippocampal neurons from NCLX-WT and NCLX-KO mice (Figure 5). In line with our previous work [25], 4 h of treatment with 10 µM XE991 of NCLX-WT hippocampal neurons produced a significant distal relocation (distance from the soma) of FGF14 along the AIS by about 4 µm, which was completely precluded by 25 µM TBCA treatment (Figure 5A–C). No change in the length of the FGF14 (AIS) segment was observed with either treatment (Figure 5C). In NCLX-KO hippocampal neurons, 4 h of chronic XE991 treatment did not produce distal relocation of FGF14, and we did not observe any effect of TBCA (Figure 5D–E and Figure 6). In these neurons, no change in AIS length was observed (Figure 5F). Interestingly, while the length of the FGF14 segment was very similar in hippocampal neurons from NCLX-WT and NCLX-KO mice, its distance from the soma was significantly shorter in NCLX-KO neurons than that in NCLX-WT mice (7.09 ± 0.41 µm and 4.87 ± 0.38 µm for untreated NCLX-WT and NCLX-KO neurons, respectively) (Figure 5G).

## 3. Discussion

The CKII metabolic role is still poorly understood. Previous studies have shown a correlation between the downregulation of CKII and mitochondrial apoptotic signaling [6]; however, the mediator of this mechanism remains unidentified. In this study, we propose that the mitochondrial Na^+^/Ca^2+^ exchanger,

NCLX, is a target of CKII phosphorylation and that CKII regulates the metabolic activity through it. Based on our following findings, phosphorylation of NCLX by CKII controls its function: (1) Mass spectral analysis of NCLX revealed a putative CKII phosphorylation site, on the Ser271 within the NCLX regulatory loop (Figure 1A,B). (2) Cells treated with CKII inhibitors displayed a much slower [Ca^2+^]m efflux, indicating an inhibited NCLX activity (Figure 1C–F). (3) It may be argued that CKII indirectly affects NCLX. To address this scenario, we generated and tested two phosphomimetic NCLX mutations, NCLX^S271D^ and NCLX^S271A^, at the CKII phosphorylation site, and found that these displayed an active and inactive NCLX-dependent mitochondrial Ca^2+^ efflux, respectively (Figure 2A–E). (4) Lastly, the activity of the mutants was insensitive to the CKII inhibitors linking the functional effect of this compound to the CKII site on NCLX (Figure 2F–H). All these findings independently indicate that the phosphorylation of the NCLX Ser271 by CKII is crucial for the regulation of this exchanger and, thereby, the mitochondrial Ca^2+^ signaling (Figure 6).

Previous studies on CKII in cancer cells have shown that it acts as a suppressor of cell apoptosis implicated by depolarization of ΔΨm [1,2,3]. Depolarization of ΔΨm is a hallmark of apoptotic signaling and cell death [43,44,45]; however, the CKII link with this metabolic function is poorly understood. NCLX is required for proper mitochondrial function [11,28,29,46]. Indeed, mitochondrial Ca^2+^ overload, following NCLX dysfunction, may lead to mitochondrial depolarization, culminating in cell death (Figure 6). Importantly, NCLX by itself is allosterically and strongly regulated by ΔΨm [12], thus creating a vicious cycle where such depolarizing effect leads to further decline in NCLX activity aggravating the mitochondrial status. Notably, our results show that when mild mitochondrial depolarization is triggered by the mild uncoupler BAM15, leading to NCLX inhibition, the expression of the phosphomimetic active NCLX^S271D^ mutation rescues mitochondrial Ca^2+^ transients. In contrast, the inactive NCLX^S271A^ mutation fails to do so (Figure 3). Hence, our results describe a novel pathway where the regulation of NCLX by CKII renders it to be less susceptible to depolarization of ΔΨm. This thereby breaks the vicious cycle where such events suppress NCLX activity, leading to mitochondrial Ca^2+^ overload and a further decline in ΔΨm. Thus, we can conclude that the phosphorylation by CKII on the NCLX Ser271 is linked to its allosteric regulation by ΔΨm. There are many reports on the regulatory effect of CKII on mitochondrial proteins primarily linked to apoptosis [22]. It is largely unknown, however, if CKII reaches and phosphorylates the substrates at the matrix or IMS. Our previous [9] and current results—suggesting that CKII does phosphorylate and modulate NCLX—provide an indication that CKII may reach indeed these domains.

How the CKII phosphorylation site is linked to the regulation of NCLX is still unclear. Nevertheless, we previously identified two clusters of positive charges located at both edges of the NCLX regulatory domain that sense changes in ΔΨm and are crucial for the allosteric regulation of NCLX by ΔΨm [9,47]. Interestingly, the CKII phosphorylation site is in proximity to these sites. Therefore, a possible scenario is that the negative charge, inserted through phosphorylation on Ser271, triggers a charge-dependent conformational change of these sites.

Further structural studies are required to gain insight into how the CKII phosphorylation is transmitted to these ΔΨm-sensing sites. The ability of protein kinases to reach the mitochondria is controversial. Although it is presently unclear how CKII reaches the mitochondria, our results indicate that it phosphorylates NCLX and modulates its activity. Therefore, NCLX may serve as a handle to interrogate the mode of CKII accessibility into this organelle. Notably, the CKII phosphorylation site is located in the regulatory domain of NCLX and in the vicinity of the PKA phosphorylation site. This domain is flanked by the catalytic α1 and α2 [9,28,29]. Therefore, it is conceivable that the phosphorylation by CKII and PKA functionally interacts and modulates this catalytic core. Future structural studies comparing, for example, the spatial organization of the phosphomimetic inactive NCLX^S271A^ and active NCLX^S271D^ may reveal their mode of interaction.

Previous studies on the fast plasticity of AIS over time, triggered by blocking axonal M-type K^+^ channels, established that it is a CKII dependent feature [25]. However, what links CKII to this homeostatic neuron process is poorly understood. Here, we asked whether this CKII-dependent AIS plasticity is linked to NCLX. Our electrophysiological data indicate that both excitatory and inhibitory hippocampal neurons derived from NCLX-KO mice exhibit higher intrinsic excitability than those from NCLX-WT mice, as reflected by the smaller threshold current (Figure 5C). NCLX is the major [Ca^2+^]m exit pathway and its inducible knock-out in adult mouse heart was shown to produce [Ca^2+^]m overload and increase reactive oxygen species (ROS), which rendered the mice cells much more susceptible to cell death upon ischemia-reperfusion [48]. Ca^2+^ overload can promote the opening of the mitochondrial transition pore (mPTP), mitochondrial swelling, cytochrome c loss, uncoupling of the oxidative phosphorylation system (OXPHOS), and excessive ROS production [49,50]. We previously found that knock-down of NCLX produces high levels of ROS, responsible for cysteine^195^ oxidation in Orai1, the plasma membrane Ca^2+^ entry channel during store-operated Ca^2+^ entry [46]. More recently, NCLX knock-down was also found to produce [Ca^2+^]m overload and to increase ROS in mouse oocytes [51].

Normal brain function depends on a close coupling between neuronal activity and ATP synthesis to prevent metabolic imbalance during increased activity levels. It is well known that mitochondrial dysfunction, such as increased ROS, can lead to hyperexcitability [52]. Elevated levels of mitochondrial ROS and dysfunction of the enzymes involved in OXPHOS are typically associated with epileptic seizures [53,54]. Increases in mitochondrial ROS was shown to trigger hyperexcitability to both mechanical and chemical stimuli in airway nociceptors [55]. Recent data showed that blocking mitochondrial OXPHOS and increased ROS levels promote hyperexcitability and epileptiform activity in rat hippocampal slices [56]. Importantly, ROS has been shown to increase the non-inactivating Na^+^ current (INaP) and decrease the voltage-gated K^+^ channel functions, notably the transient A-type and the delayed rectifying K^+^ current, which can significantly contribute to network hyperexcitability [52,57,58,59]. Along the same line of enquiry, a recent study showed that deficient oxidative phosphorylation in a mouse strain carrying a conditional deletion of MPC1, an essential subunit of the mitochondrial pyruvate carrier, developed seizures in response to mild inhibition of GABA-mediated synaptic activity [60]. Neurons from these deficient MPC1 mice were intrinsically hyperexcitable as a consequence of impaired calcium homeostasis, which reduced the M-type potassium channel activity [60]. Thus, it is possible that the intrinsic hyperexcitability found in hippocampal neurons derived from NCLX-KO mice could arise from reduced activity of voltage-gated K^+^ currents.

The intrinsic hyperexcitability observed in this study is in line with our immune cyto-fluorescence results, which showed that the distance from the soma of the FGF14 segment was significantly shorter in NCLX-KO neurons than that in NCLX-WT mice (Figure 5E). A previous study revealed that activity-dependent distal relocation of the AIS away from the soma of cultured hippocampal neurons causes a reduction in intrinsic excitability [61]. It was proposed that a greater passive attenuation of the voltage transients on the way from the soma to the AIS trigger zone enriched in Na^+^ channels is responsible for the reduction in excitability [61]. Using compartmental simulation models [25], we previously showed that distal relocation of the AIS containing both Na^+^ and Kv7.2/3 K^+^ channels (M-channels) causes a decrease in excitability [25]. Thus, within the AIS, the more distally located Kv7 channels are more effective in suppressing neuronal excitability than those located more proximally. In the present work, the FGF14 segment, which bridges Na^+^ and Kv7 channels, is significantly shorter in NCLX-KO mice, which is in line with an increase in intrinsic excitability of hippocampal NCLX-KO neurons as compared to those from NCLX-WT mice.

Our electrophysiological data indicate that chronic M-channel block with XE991 triggered homeostatic adaptation with a return of the threshold current to initial untreated values for both excitatory and inhibitory NCLX-KO hippocampal neurons (Figure 4D). Thus, as with NCLX-WT neurons, NCLX-KO neurons can exhibit homeostatic plasticity following M-channel block. However, at the mechanistic level, NCLX-KO neurons do not use a CKII-sensitive distal relocation of AIS Na^+^ or Kv7 channels to decrease their intrinsic excitability, as WT neurons do (Figure 6). NCLX-KO hippocampal neurons may utilize a different compensatory mechanism from NCLX-WT neurons, such as by modulating the density of voltage-gated Na^+^ and/or K^+^ channels. In other words, the presence of mitochondrial NCLX is essential for producing the homeostatic distal relocation of AIS Na^+^ and Kv7 channels triggered by M-channel block in NCLX-WT hippocampal neurons.

## 4. Materials and Methods

### 4.1. Primary Neuronal Culture and Infection

C57BL6 Wild-Type (WT) (Envigo, Indianapolis, Indiana, C57BL Cat #6JRCCH5B043) and NCLX-KO mice were obtained from Jackson Laboratories (Bar Harbor, Maine, USA, Cat #026242,) and grown at the Ben-Gurion University mouse facility. This strain was generated at the Jackson Laboratory by injecting Cas9 RNA and an 18-mer guide sequence ATACTGGAGACGGCGTCT. This resulted in a 13bp deletion (AGACGGCGTCTGG) in exon 2 beginning at chromosome 5 positive strand position 120, 513, 241bp (GRCm38), which is predicted to cause amino acid sequence changes after residue 21 and an early truncation 33 amino acids late.

Animals were treated with the approval and in accordance with the guidelines of the Ben-Gurion University Institutional Committee for Ethical Care and Use of Animals (Reference Number: IL 07-02-2019C).

Primary culture of hippocampal neurons was done for each independent experiment from mouse pups as previously described [30]. Each mouse hippocampus was used to generate cultures on six coverslips with 50,000–70,000 cells. Cultures were typically maintained at 37 °C in a 5% CO_2_ humidified incubator for 10–15 days before experimentation.

Primary hippocampal cultures from NCLX-WT and NCLX-KO mice were infected with adeno-associated viral particles (AAV) carrying cDNAs of proteins of interest or fluorescent reporter 2MT-GCaMP6m (mitoGCaMP6m). Viral particles were produced in HEK293T cells using both the pD1 and pD2 helper plasmids, which encode the rep/cap proteins of AAV1 and AAV2, respectively. Primary cultures of hippocampal neurons were infected at 5 DIV and incubated for at least 7 days before imaging. The virus titer was set to produce 75–90% infection efficiency.

### 4.2. Cell Cultures and Transfection

SH-SY5Y and HEK293-T cell lines were cultured (37 °C, 5% CO_2_) in Dulbecco’s modified Eagle medium (DMEM) supplemented with 10% fetal bovine serum (FBS), 2mM L-glutamine, and 1% penicillin/streptomycin, as previously described [34]. Transfection of cells was performed using the calcium phosphate precipitation protocol as previously described [34]

### 4.3. Live Fluorescence Imaging

Experiments were done on an imaging system consisting of Olympus IX73 inverted microscope, Cellsense division 1 Olympus software (Wendenstrasse, Hamburg, Germany), CooLed 4000 LED monochromator, and Q imaging cooled Retiga 6000 camera (Surrey, British Columbia, Canada). We used a magnification of 20× for mitochondrial measurements (Cepia2mt and TMRM), and 10× for live cytosolic Ca^2+^ (Fluo-4) imaging as previously described [12].

Experiments done on NCLX-WT and NCLX-KO live neurons stimulated electrically were performed using the imaging system consisting of a Nikon TiE inverted microscope driven by the NIS-elements software package (Nikon). The microscope was equipped with an Andor sCMOS camera (Oxford Instruments), a 40× 0.75 NA Super Fluor objective, a 60× 1.4 NA oil-immersion apochromatic objective (Nikon), a perfect-focus mechanism (Nikon), and EGFP, EYFP, and Cy3 TE-series optical filter sets (Chroma) as well as BFP and Cy5 filter sets (Semrock) [30]. The baseline fluorescence intensity of MitoGcaMP6m (F0) in each area of interest was the calculated average value measured in 10 successive images acquired before stimulation. The change in fluorescence (ΔF) to time (t) was calculated as (F(t)-F0)/F0.

At the beginning of each experiment, cells were perfused with Ringer’s solution containing (in mM): 126 NaCl, 5.4 KCl (Sigma-Aldrich, Darmstadt, Germany, Cat #P9333), 0.8 MgCl_2_, 20 HEPES, 15 glucose (Gerbu, Heidelberg, Germany, Cat #2028), with pH adjusted to 7.4 with NaOH (Sigma-Aldrich, Darmstadt, Germany Cat #221465) and 1.8 CaCl_2_. To trigger ionic responses in SH-SY5Y, cells were perfused with Ringer’s solution containing ATP (100 μM) (Amresco, Ohio, USA Cat # 0220).

### 4.4. Western Blotting

For protein determination from whole-cell lysates, cells were first washed with ice-cold PBS and lysed for 30 min on ice with RIPA buffer (Sodium pyrophosphate 10 mM, Tritone X-100 2%, NaCl 100 mM, 5 mM EDTA pH 7.4, 5 mM EGTA pH 7.4, deoxycholate 0.5%, NaF 25 mM, and Na orthovanadate 1 mM). Lysates were centrifuged at 4 °C for 20 min at 12,000 rpm, and the supernatant was collected. For the determination of protein expression in mitochondria, cell fractionation was performed as previously described [9]. Protein concentrations were determined using the Bradford assay (Bio-Rad). Equal protein quantities were subjected to SDS-PAGE gel electrophoresis. Immunoblot analysis was performed using an anti-myc antibody (1:1000) TBST/5% skim milk, followed by incubation with a secondary anti-rabbit-IgG antibody coupled to HRP (1:5000). Detection was done with the EZ-ECL Chemiluminescence Detection kit for HRP (Biological Industries) on an ImageQuant LAS 4000 digital imaging system (General Electric Healthcare Bio-Sciences).

### 4.5. Plasmids

For the generation of NCLX point-mutants, site-directed mutagenesis was performed on the NCLX double-stranded plasmid (containing the NCLX gene; accession number AY601759.1), using the QuikChangesite-directed mutagenesis kit (Stratagene, Santa Clara, California, USA) according to protocols provided by the manufacturer. The following primers were used:NCLX^S271A^ forward: GTTACTCCAGAGATCCTCGCAGACTCCGAG;NCLX^S271A^ reverse: CTCGGAGTCTGCGAGGATCTCTGGAGTAAC;NCLX^S271D^ forward: CTCCAGAGATCCTCGATGACTCCGAGGAGGAC;NCLX^S271D^ reverse: GTCCTCCTCGGAGTCATCGAGGATCTCTGGAG.

The human NCLX shRNA plasmid was obtained from Sigma-Aldrich (Darmstadt, Germany, Mission TRC shRNA Target Set TRCN-5045). Plasmid pcDNA3.1+ was from Thermo Fisher (Waltham, Massachusetts, USA, Cat# V79020). Plasmid encoding human NCLX tagged with c-myc at the C-terminal was obtained from OriGene (Givat Brenner, Israel, Cat#RC214624).

### 4.6. [Ca^2+^]m Measurements

Where stated, cells were incubated with drugs prior to the [Ca^2+^]m recording. The following conditions were used: BAM15 (5 μM for 15 min), TBB (10 μM for 4 h), TBCA (10 μM for 2 h).

[Ca^2+^]m levels were monitored in SH-SY5Y cells by transiently-expressed mitochondrial-targeted [Ca^2+^]m sensor-Cepia2mt. Cepia2mt fluorescence was acquired at 480 nm and collected through a 535 nm band-pass filter. Cells were initially perfused with Ca^2+^-full Ringer’s solution (1.8 mM CaCl_2_, 120 mM NaCl, 5.4 mM KCl, 0.8 mM MgCl_2_, 20 mM HEPES, 15 mM glucose, and pH adjusted to 7.4 with NaOH) to record baseline signal. [Ca^2+^]m response was triggered by switching the perfusion solution to Ca^2+^-free Ringer’s solution supplemented with 100 μM ATP, as previously described [28].

### 4.7. Cytosolic Ca^2+^Measurements

Where stated, cells were incubated with TBCA prior to the Ca^2+^ recording. The following conditions were used: TBCA (10 μM for 2 h).

Cytosolic Ca^2+^ levels were monitored in SH-SY5Y preloaded with Fluo-4. Cells were initially preloaded with Fluo-4 (2 µM) for 25 min in Ca^2+^-full Ringer’s solution (see [Ca^2+^]m measurements) in RT, then the cells were washed for 15 min in Ca^2+^-full Ringer’s solution in RT. Fluo-4 fluorescence was acquired at 480 nm and collected through a 535 nm band-pass filter. Cytosolic Ca^2+^ response was triggered by switching the perfusion solution from Ringer’s solution to Ca^2+^-free Ringer’s solution supplemented with 100 μM ATP, as previously described [28].

### 4.8. ΔΨm Measurements

TMRM was used in non-quenching mode to measure pre-existing ΔΨm. TMRM was excited at 545 nm and imaged with a 570 nm emission filter as previously described [62]. Cells were initially preloaded with TMRM (30 μM) Ca^2+^-full Ringer’s solution (see [Ca^2+^]m measurements) for 30 min in 37 °C, then the cells were washed with TMRM (10 μM) for 15 min in RT in Ca^2+^-full Ringer’s solution. TMRM was added to all experimental solutions. To trigger complete depolarization (Fmin), 5 μM FCCP was added to the cells. ΔΨm was calculated as F-Fmin, where F was the fluorescence signal at each time point, and Fmin was the fluorescence of complete depolarization.

### 4.9. Drugs

XE991 dihydrochloride (Tocris 2000/10), Bicuculline Methiodide (Alomone B-136), D-2-Amino-5-phosphonovaleric acid (AP-5, Alomone D-145), NBQX disodium salt (Alomone N-186), tetrabromocinnamic acid (TBCA) (Sigma-Aldrich, Darmstadt, Germany), NBQX (Sigma-Aldrich, Darmstadt, Germany), AP5 (Sigma-Aldrich, Darmstadt, Germany), 4,5,6,7-tetrabromobenzotriazole (TBB) (Cayman chemical company, Ann Arbor, Michigan, USA), N5,N6-bis(2-Fluorophenyl)[1,2,5]oxadiazolo[3,4-b]pyrazine-5,6-diamine (BAM15) (Cayman chemical company, Ann Arbor, Michigan, USA)

### 4.10. Primary Cultures of Hippocampal Neurons

Hippocampi were dissected from neonate C57BL6mice brains (0–1 days old). Hippocampi were washed three times in an HBSS-based solution containing: 4 mM NaHCO_3_, 5 mM HEPES, and Hank’s balanced salt solution (Sigma-Aldrich, Darmstadt, Germany), and pH adjusted to 7.3–7.4 at 4 °C. Tissues were digested in a solution including: 137 mM NaCl, 5 mM KCl, 7 mM Na_2_HPO_4_, 25 mM HEPES, 4.45 mg/mL trypsin type XI (Sigma-Aldrich, Darmstadt, Germany) and 1614 U/mL DNase type IV (Sigma-Aldrich, Darmstadt, Germany), with pH adjusted to 7.2 at 4 °C. Hippocampal tissues were incubated for 10 min at 37 °C and washed once with 5 mL HBSS/20% FBS (fetal bovine serum) and once with HBSS. The cells were dissociated in an HBSS solution including 13.15 mM MgSO4 and 1772 U/mL DNase type IV (Sigma-Aldrich, Darmstadt, Germany). Next, the cells were mechanically triturated with fire-polished Pasteur pipettes. HBSS/20% FBS was added to the dissociated cells, and the mixture was centrifuged at 1000× *g* at 4 °C for 10 min. The supernatant was discarded and a plating medium including MEM (Thermo Fisher, Waltham, MA, USA), 24.7 mM glucose, 0.089 mg/mL transferrin (Calbiochem, San Diego, CA, USA), glutaMAX (Thermo Fisher, Waltham, MA, USA), 0.75 U/mL insulin (Sigma-Aldrich, Darmstadt, Germany), 10% FBS (Biological Industries, Kibbutz Beit-Haemek, Israel) and SM1 (StemCellNeuroCult neuronal supplement) was added to the pellet. The cells were resuspended in the plating medium with a fire-polished Pasteur pipette, and viable cells were counted; 0.5 mL of the cell suspension was added to glass coverslips coated with Matrigel (Corning) in a 24-wells plate at a density of ~180,000 cells per well. Two days after plating, 0.5 mL of feeding medium (MEM, 26.92 mM glucose, 0.097 mg/mL transferrin, glutaMAX, SM1, and 3 μM cytosine arabinoside [Ara-C] (Sigma-Aldrich, Darmstadt, Germany) was added to each well. Twice a week, half of the medium was removed from the wells and replaced with the same volume of the feeding medium.

### 4.11. Recombinant AAV-Dlx-mCherry Plasmid and Infection

To identify GABAergic neurons, we infected hippocampal cultures with a recombinant virus derived from an AAV-viral vector driving the expression of the fluorescent protein mCherry under the control of the specific GABAergic hDlx promoter. Recombinant AAV-virus-Dlx-mCherry plasmid was prepared by inserting the hDlx promoter sequence (541 bp) upstream of the coding sequence in the backbone of the pAAV2-mCherry plasmid. The Dlx promoter was shown to restrict reporter expression in vivo to all GABAergic interneurons in the forebrain, including the hippocampus, as well as in cultured neurons in vitro [23,41]. The recombinant AAV2-virus-Dlx-mCherry was produced using standard production methods in HEK 293 cells. All batches produced were in the range of 109 to 1010 viral particles per ml. Infections of hippocampal cultures were performed at 6 DIV, and recordings were carried out at 12–14 DIV.

### 4.12. Patch-Clamp Electrophysiology

Patch clamp was performed in the whole-cell configuration. Signals were filtered at 4 kHz and digitized at 10 kHz. All signals were amplified using Multiclamp 700B (Molecular Devices, San Jose, CA, USA). For recordings in the current-clamp configuration, the extracellular solution contained 145 mM NaCl, 3 mM KCl, 10 mM HEPES, 15 mM glucose, 1.2 mM MgCl_2_ and 1.8 mM CaCl_2_ (pH was adjusted to 7.4 with NaOH, osmolarity ~315 mOsm). Microelectrodes with resistances of 4–7 MΩ were pulled from borosilicate glass capillaries (Harvard Apparatus, Holliston, MA, USA) and filled with an intracellular solution. The intracellular solution contained: 135 mM KCl, 1 mM KATP, 1 mM MgATP, 2 mM EGTA, 1.1 mM CaCl_2_, 10 mM HEPES and 5 mM glucose (pH adjusted to 7.3 with KOH, osmolarity ~300 mOsm). For the experiments inspecting the intrinsic properties of the neurons, synaptic blockers were added to the extracellular solution to prevent spontaneous spikes: 30 μM picrotoxin, 10 μM Bicuculline, 10 μM NBQX, and 10 μM AP-5.

### 4.13. Immunostaining

Neurons were fixed in 4% paraformaldehyde for 10 min and washed four times with phosphate-buffered saline (PBS). Permeabilization of the membrane was performed by adding 0.1% Triton X-100 (Sigma-Aldrich, Darmstadt, Germany) in a blocking solution (PBS with 0.1% BSA and 5% goat serum) for 4 min. After washing once with PBS, a blocking solution was added to the coverslips for 15 min. Primary antibodies were added, and the neurons were incubated for one hour at room temperature. The coverslips were washed three times with PBS and incubated at room temperature for one hour with the secondary antibodies. After washing three times with PBS, the coverslips were mounted in Fluoromount (Sigma-Aldrich, Darmstadt, Germany). The primary antibodies used for immunostaining were: mouse α-FGF14 (1:500; clone N56/21, catalog #75 096, lot 413-8RR-61, NeuroMab, Davis, CA, USA); and rabbit α-MAP2 (1:1000; catalog #AB5622, lot 2795016, Millipore, Burlington, MA, USA). The secondary antibodies used were donkey α-mouse Alexa488 (1:1000, Jackson, Lansing, MI, USA, 715-545-150) and donkey α-rabbit Cy3 (1:1000, Jackson, Lansing, MI, USA, 711-165-152).

### 4.14. Antibodies

In Table 1 are presented all antibodies used in this study.

### 4.15. Data Analysis and Statistics

For all experiments, data were collected from at least three different batches. Control cells were collected from each batch to minimize possible variations between batches. All graphs were built with Prism 9.0 (GraphPad, San Diego, CA, USA). Error bars represent standard error of the mean (SEM). In acute RTG treatments that were carried out in the same neuron, statistical comparisons between untreated and treated cells were performed using a two-tailed paired *t*-test. When chronic treatments involved more than two independent groups of cells without matching between measures, statistical comparisons were performed using one-way ANOVA and, post-hoc, Dunnett’s multiple comparison test. For evoked spike discharge, statistical comparisons were performed using two-way ANOVA and Bonferonni correction. Analyses of patch-clamp recordings were performed with Clampfit 10.4 (Molecular Devices, San Jose, CA, USA). Images from immunocytochemistry experiments were obtained using confocal microscopy STED TCS SP5 II microscope (Leica, Wetzlar, Germany) with oil-immersion objectives of 63×. Images were converted to TIFF files and imported into MATLAB R2021a (Natick, MA, USA) for blind analysis using a self-written algorithm. Semi-automatically, the algorithm recognizes the soma and the axon. At each pixel along the axon, fluorescence intensity is measured. The AIS was defined to be between the proximal and distal points along the axons, in which the fluorescence intensity was above 50% of the maximum. The AIS length and distance from the soma were then calculated and exported to a Microsoft Excel 2016 file (Redmond, WA, USA).

## Figures and Tables

**Figure 1 cells-11-03990-f001:**
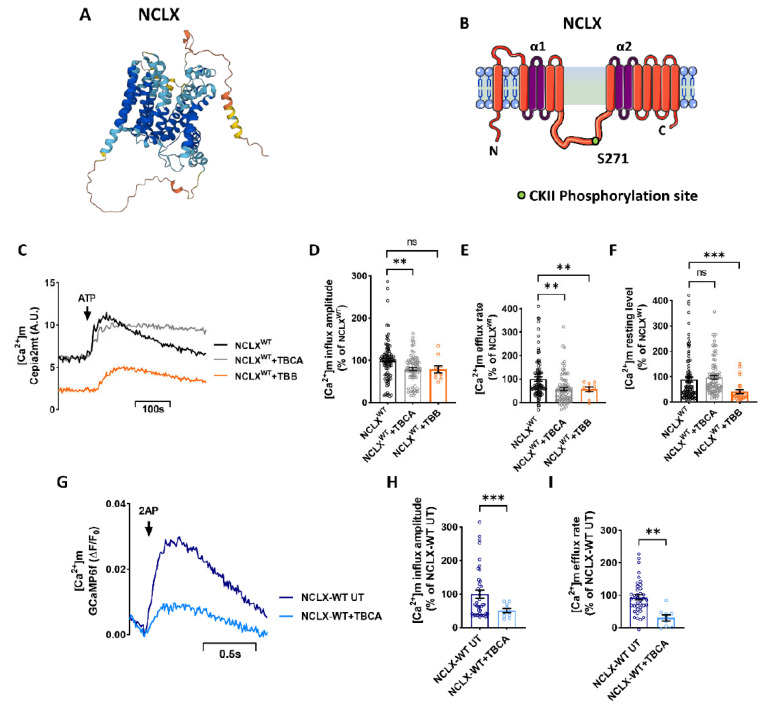
Inhibition of CKII by TBB and TBCA inhibits the [Ca^2+^]m transient. (**A**,**B**) Schematic model of NCLX membrane organization. (**A**) Was provided by AlphaFold Protein Structure Database. (**B**) The NCLX regulatory loop is located between the catalytic α1 and α2 domains. CKII phosphorylation site S271 is marked. (**C**) Representative SH-SY5Y cells co-expressing the [Ca^2+^]m fluorescent sensor Cepia2mt, shNCLX and NCLX^WT^ (black), treated with TBCA (10 μM) for 2 h (grey), or TBB (10 μM) for 4 h (orange). The [Ca^2+^]m transient was triggered by application of extracellular ATP (100 μM) in a Ca^2+^-free Ringer’s solution. A.U.—arbitrary units. (**D**) Quantification of [Ca^2+^]m influx amplitudes in (**C**) (Welch and Brown-Forsythe ANOVA, F(2,96.56)= 6.538; Dunnett’s T3 multiple comparisons test; NCLX^WT^ vs. NCLX^WT^+TBCA: ***p* = 0.0082; NCLX^WT^ vs. NCLX^WT^+TBB: ns; *n* > 3). (**E**) Quantification of [Ca^2+^]m efflux rates in (**C**) (Welch and Brown-Forsythe ANOVA, F(2,151.4)= 11.3; Dunnett’s T3 multiple comparisons test; NCLX^WT^ vs. NCLX^WT^+TBCA: ***p* = 0.0014; NCLX^WT^ vs. NCLX^WT^+TBB; *n* > 3). (**F**) Quantification of [Ca^2+^]m basal intensity in (**C**) (Welch and Brown-Forsythe ANOVA, F (2,163.4) = 5.252; Dunnett’s T3 multiple comparisons test; NCLX^WT^ vs. NCLX^WT^+TBCA: ns; NCLX^WT^ vs. NCLX^WT^+TBB: ****p* = 0.0009; *n* > 3). (**G**) Representative fluorescent traces action-potential (AP) induced influx of GCaMP6f expressing NCLX-WT hippocampal neurons derived from C57BL6 mice, treated (light blue) and untreated (deep blue) with TBCA (25 μM) for 2 h, stimulated at 2 Hz for 1.3 s. (**H**) Quantification of [Ca^2+^]m influx amplitude in (**G**) (unpaired two-tailed *t*-test, *t* = 3.580, df = 46.90, ****p* = 0.0008; *n* > 3). (**I**) Quantification of [Ca^2+^]m efflux rates in (**G**) (unpaired two-tailed *t*-test, *t* = 2.21, df = 49, *p* = 0.0318; *n* > 3). *n* > 3. ns—not significant. Error bars denote SEM.

**Figure 2 cells-11-03990-f002:**
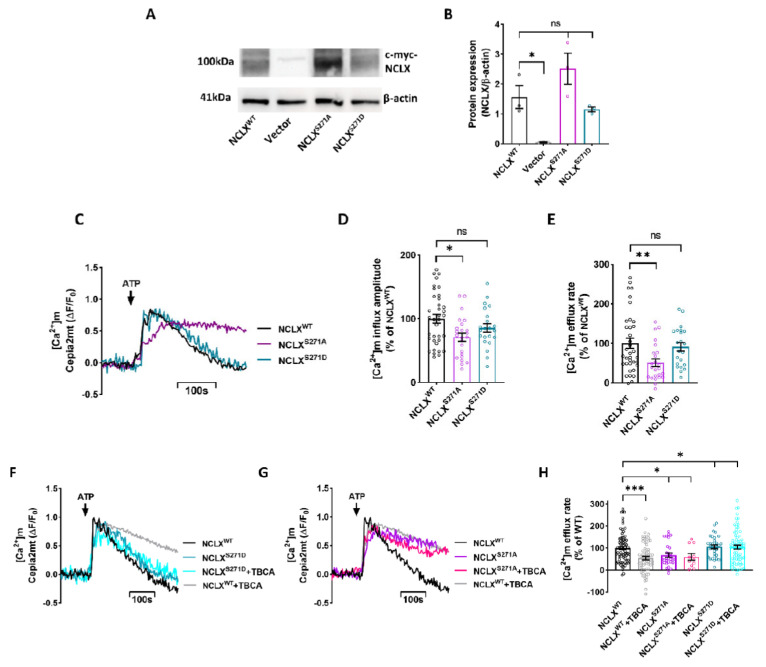
NCLX activity is controlled by phosphorylation on Ser271. (**A**) Representative immunoblot analysis (Western blot) of NCLX c-myc-tagged constructs (For Antibodies see Table 1). Extracts of HEK293 cells transfected with shNCLX and either the NCLX^WT^, NCLX^S271A^, NCLX^S271D^ or control vector pcDNA3.1+ (vector) were subjected to immunoblot analysis with c-Myc and β-Actin antibodies. (**B**) Densitometry analysis of the Western blot shown in (**A**). Protein expression was normalized to β-Actin expression (unpaired two-tailed *t*-test, NCLX^WT^ vs. Vector: *t* = 3.979, df = 4, **p* = 0.0164; NCLX^WT^ vs. NCLX^S271A^: *t* = 1.469, df = 4ns; NCLX^WT^ vs. NCLX^S271D^: *t* = 1.055, df = 4, ns; *n* = 3). (**C**) Representative fluorescent traces of ATP (100 μM) induced [Ca^2+^]m transients in Cepia2mt expressing SH-SY5Y cells co-expressing shNCLX and NCLX^WT^ (black) or mutant NCLX^S271A^ (purple)\NCLX^S271D^ (turquoise). (**D**) Quantification of [Ca^2+^]m influx amplitude in (**C**) (Welch and Brown-Forsythe ANOVA, F(2,78.46) = 5.028; Dunnett’s T3 multiple comparisons test; NCLX^WT^ vs. NCLX^S271A^: **p* = 0.011; NCLX^WT^ vs. NCLX^S271D^: ns; *n* > 3). (**E**) Quantification of [Ca^2+^]m efflux rates in (**C**) (Welch and Brown-Forsythe ANOVA, F(2,77.72) = 5.519; Dunnett’s T3 multiple comparisons test; NCLX^WT^ vs. NCLX^S271A^: ***p* = 0.0091; NCLX^WT^ vs. NCLX^S271D^: ns; *n* > 3). (**F,G**) Representative fluorescent traces of ATP (100μM) induced [Ca^2+^]m transients in Cepia2mt expressing SH-SY5Y cells co-expressing shNCLX and NCLX^WT^ (black or grey with TBCA) or mutant NCLX^S271A^ (purple or pink with TBCA)\NCLX^S271D^ (turquoise or light blue with TBCA) treated with TBCA. (**H**) Quantification of [Ca^2+^]m efflux rates in (**G**,**H**) (Welch and Brown-Forsythe ANOVA, F(5,220.7) = 7.435; Dunnett’s T3 multiple comparisons test; NCLX^WT^ vs. NCLX^WT^+TBCA: ****p* = 0.0006; NCLX^WT^ vs. NCLX^S271A^: **p* = 0.0126; NCLX^WT^ vs. NCLX^S271A^+TBCA: **p* = 0.0292; NCLX^WT^ vs. NCLX^S271D^: ns; NCLX^WT^ vs. NCLX^S271D^+TBCA: ns; *n* > 3). ns—not significant. Error bars denote SEM.

**Figure 3 cells-11-03990-f003:**
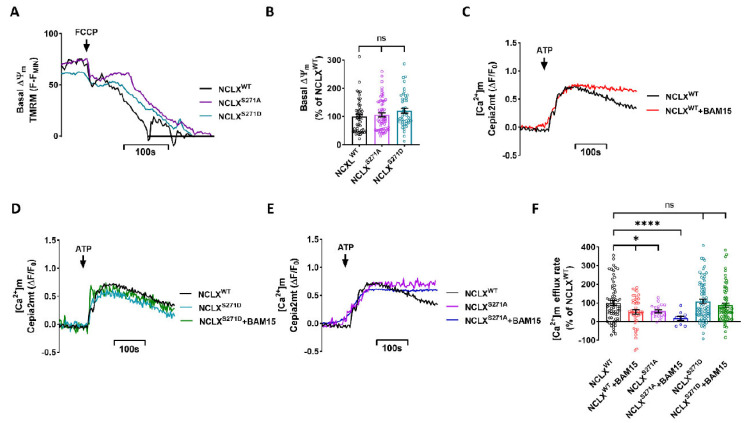
Mild mitochondrial depolarization, triggered by chemical uncoupling, allosterically inhibits [Ca2+]m efflux by NCLX and is prevented by phosphomimetic mutant of NCLX^S271D^. (**A**) Representative fluorescent traces of ΔΨm on SH-SY5Y cells co-expressing shNCLX and NCLX^WT^ (black) or mutant NCLX^S271A^ (purple)\NCLX^S271D^ (turquoise) preloaded with TMRM (30 μM for loading and 10 μM for washing in all experimental solutions). FCCP (5 μM) was added to calibrate the signal by inducing complete depolarization, observed as a drop in TMRM fluorescence. (**B**) Quantification of basal ΔΨm in (**A**) (Welch and Brown-Forsythe ANOVA, F(2,135.3) = 1.414;Dunnett’s T3 multiple comparisons test; NCLX^WT^ vs. NCLX^S271A^: ns; NCLX^WT^ vs. NCLX^S271D^: ns; *n* > 3). (**C**–**E**) Representative fluorescent traces of ATP (100 μM) induced [Ca^2+^]m transients in Cepia2mt expressing SH-SY5Y cells co-expressing shNCLX and NCLX^WT^ (black or red with BAM15), or mutant NCLX^S271A^ (purple or blue with BAM15)\NCLX^S271D^ (turquoise or green with BAM15) treated and untreated with BAM15 (5 μM) for 15 min. (**F**) Quantification of [Ca^2+^]m efflux rates (C–E) (Welch and Brown-Forsythe ANOVA, F(5,279.1) = 4.762; Dunnett’s T3 multiple comparisons test; NCLX^WT^ vs. NCLX^WT^+BAM15: **p* = 0.0380; NCLX^WT^ vs. NCLX^S271A^: **p* = 0.0195; NCLX^WT^ vs. NCLX^S271A^+BAM15: *****p* > 0.0001; NCLX^WT^ vs. NCLX^S271D^: ns; NCLX^WT^ vs. NCLX^S271D^+BAM15: ns; *n* > 3). ns—not significant. Error bars denote SEM.

**Figure 4 cells-11-03990-f004:**
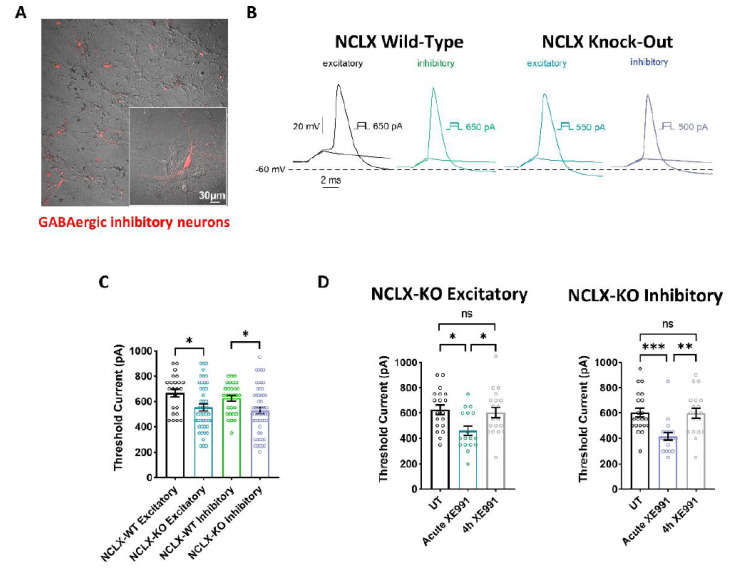
Intrinsic excitability properties of hippocampal neurons derived from NCLX-WT and NCLX-KO mice. (**A**) Cultured hippocampal neurons infected with recombinant AAV vector driving the expression of the fluorescent protein mCherry, under the control of the specific GABAergic mDlx enhancer. GABAergic inhibitory neurons expressing mCherry are seen as red. (**B**) Representative traces of solitary spike discharge evoked by 2 ms step injection of depolarizing currents with increments of 50 pA in NCLX-WT excitatory neurons (black), NCLX-WT inhibitory neurons (green), NCLX-KO excitatory neurons (blue), and NCLX-KO inhibitory neurons (purple). (**C**) Threshold current for spike firing of NCLX-KO compared to NCLX-WT excitatory and inhibitory neurons (one-way ANOVA, F(3,150) = 4.793; Bonferroni’s multiple-comparisons test; NCLX-WT excitatory vs. NCLX-KO excitatory: **p* = 0.0176; NCLX-WT inhibitory vs. NCLX-KO inhibitory: **p* = 0.0268; *n* > 3). (**D**) Threshold currents of untreated NCLX-KO excitatory (left) and NCLX-KO inhibitory (right) neurons, compared to neurons acutely exposed to 10 μM XE991 and after 4 h chronic exposure (excitatory: one-way ANOVA, F(2,52) = 5.251; Tukey’s multiple-comparisons test; UT vs. acute XE991: **p* = 0.0108; Acute XE991 vs. 4 h XE991: **p* = 0.0325; UT vs. 4 h XE991: ns; Inhibitory: one-way ANOVA, F(2,56) = 9.277; Tukey’s multiple-comparisons test; UT vs. acute XE991: ****p* = 0.0010; Acute XE991 vs. 4 h XE991: ***p* = 0.0018; UT vs. 4 h XE991: ns; *n* > 3). ns—not significant. Error bars denote SEM.

**Figure 5 cells-11-03990-f005:**
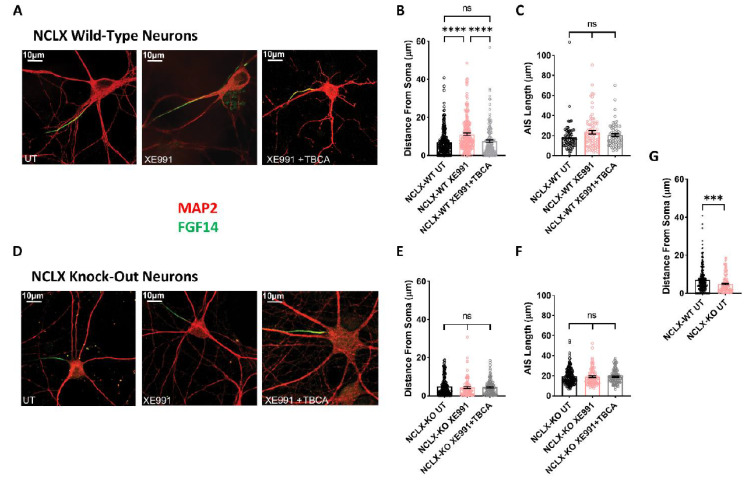
Plasticity of the axon initial segment in hippocampal neurons derived from NCLX-WT and NCLX-KO mice. (**A**) Representative FGF14 (green) and MAP2 (red) immunostaining of NCLX-WT hippocampal neurons (For Antibodies see Table 1). (**B**) In NCLX-WT hippocampal neurons, the AIS relocated distally away from the soma following 4 h of exposure to XE991, but the addition of TBCA prevented this effect (one-way ANOVA, F(2,606) = 16.89; Tukey’s multiple-comparisons test; NCLX-WT UT vs. NCLX-WT+XE991: *****p* < 0.0001; NCLX-WT+XE991 vs. NCLX-WT+XE991+TBCA: *****p* < 0.0001; NCLX-WT UT vs. NCLX-WT+XE991+TBCA: ns; n > 3). (**C**) No significant differences were observed in AIS length (one-way ANOVA, F(2,214) = 2.225; Tukey’s multiple-comparisons test; NCLX-WT UT vs. XE991: ns; NCLX-WT+XE991 vs. NCLX-WT+XE991+TBCA: ns; n > 3). (**D**) Representative FGF14 (green) and MAP2 (red) immunostaining of NCLX-KO hippocampal neurons. (**E**) In NCLX-KO hippocampal neurons, no significant difference in AIS distance from the soma was observed (one-way ANOVA, F(2,346) = 0.3239; Tukey’s multiple-comparisons test; NCLX-KO UT vs. XE991: ns; NCLX-KO+XE991 vs. NCLX-KO+XE991+TBCA: ns; NCLX-KO UT vs. NCLX-KO+XE991+TBCA: ns; *n* > 3). (**F**) No difference in AIS length was observed (one-way ANOVA, F(2,346) = 0.1053; Tukey’s multiple-comparisons test; NCLX-KO UT vs. XE991: ns; NCLX-KO+XE991 vs. NCLX-KO+XE991+TBCA: ns; *n* > 3). (**G**) NCLX-KO hippocampal neurons exhibited a significantly shorter AIS distance from the soma compared to NCLX-WT hippocampal neurons (unpaired two-tailed *t*-test, *t* = 3.626, df = 394, ****p* = 0.0003; *n* > 3). ns—not significant. Error bars denote SEM.

**Figure 6 cells-11-03990-f006:**
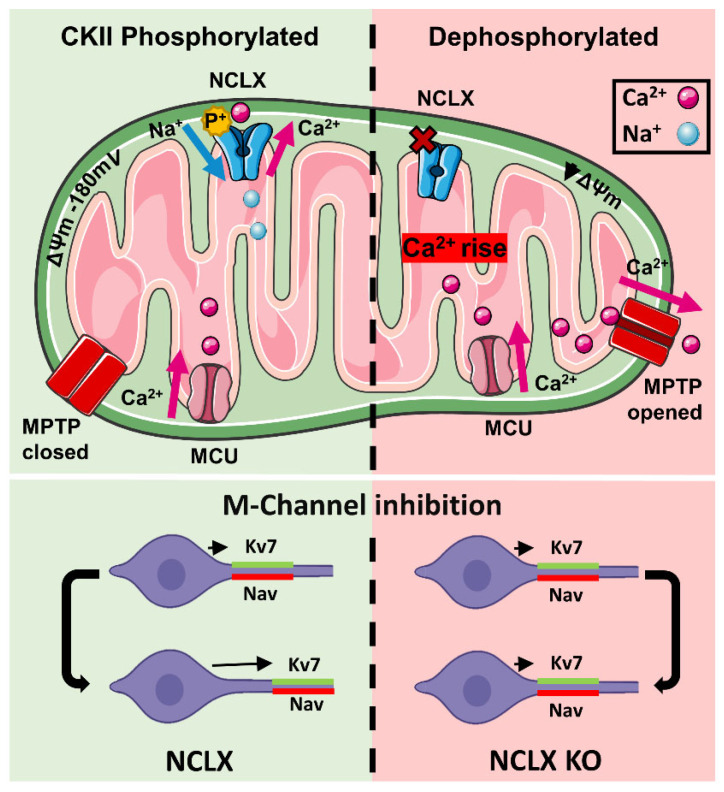
Regulation pathway of NCLX by CKII and its implication in controlling neuronal plasticity of the axon initial Scheme 2. efflux. CKII phosphorylation also participates in the allosteric regulation of NCLX-dependent Ca^2+^ transport by ΔΨm. The link between CKII and NCLX is critical for controlling ultrafast AIS plasticity. Knocking down of NCLX eliminates the physical and functional translocation of the axon initial segment linked to this process.

**Table 1 cells-11-03990-t001:** Antibodies.

Antibodies	Manufacturer	Catalog Number
Mouse monoclonal anti-Myc tag	Abcam	Cat# AB18185
Mouse monoclonal anti-β-actin	Sigma-Aldrich	Cat#A2228; RRID:AB_476697
Goat anti-mouse IgG	Jackson	Cat# 115-035-146; RRID:AB_2307392
Mouse a-FGF14	NeuroMab	Cat#75 096
Rabbit a-MAP2	Millipore	Cat#AB5622
Donkey a-mouse Alexa488	Jackson	Cat#715- 545-150
Donkey a-rabbit Cy3	Jackson	Cat#711-165-152

## Data Availability

The data presented in this study is available on request from the corresponding author.

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
