# Peer review of "CKII Control of Axonal Plasticity Is Mediated by Mitochondrial Ca2+ via Mitochondrial NCLX"

_cells, 2022, doi:10.3390/cells11243990_

Round 1

Reviewer 1 Report

In this manuscript, Sekler’s group showed that NCLX was regulated by Casein Kinase 2 (CKII) at the serine 271 in neuronal cells. Furthermore, they identify NCLX as a crucial link between CKII signaling and fast neuronal plasticity. In fact, studying the immunolocalization of FGF14, a protein physically bridging Nav1.6 and Kv7.2 channels along the AIS, the authors showed that treatment with the inhibitor of M-current XE991 produced a significant distal relocation (distance from the soma) of FGF14 along the AIS in WT hippocampal neurons. This effect was completely inhibited by the CKII inhibitor TBCA. In addition the distance of FGF14 from the soma was significantly shorter in NCLX-KO neurons than that in WT mice.

This is an interesting study that sheds light on a very important mechanism for neuronal excitability.

 Moreover, some clarification should be done.

1)For instance the authors should better clarify the effect of CKII inhibitor TBCA on mitochondrial influx in hyppocampal neurons stimulated at 2 Hz for 1.3 s to generate an action potential. They call into question the involvement of mitochondrial MCU, but do not give direct evidence of this involvement.

2)Another question concerns the effect of CKII blockade on global [Ca2+]i. Possibly some experiment should be done to demonstrate the specificity of TBCA on NCLX activity and [Ca2+]m.

In this respect, in the paragraph “Live fluorescence imaging” the use of Fura2 is reported but any experiment was shown.

Also for culture cells, experiments in HEK293 should be added, as reported in the “cell culture” paragraph.

Author Response

Thank you for the helpful and constructive suggestion that we fully addressed :

Comments and Suggestions for Authors

In this manuscript, Sekler’s group showed that NCLX was regulated by Casein Kinase 2 (CKII) at the serine 271 in neuronal cells. Furthermore, they identify NCLX as a crucial link between CKII signaling and fast neuronal plasticity. In fact, studying the immunolocalization of FGF14, a protein physically bridging Nav1.6 and Kv7.2 channels along the AIS, the authors showed that treatment with the inhibitor of M-current XE991 produced a significant distal relocation (distance from the soma) of FGF14 along the AIS in WT hippocampal neurons. This effect was completely inhibited by the CKII inhibitor TBCA. In addition the distance of FGF14 from the soma was significantly shorter in NCLX-KO neurons than that in WT mice.

This is an interesting study that sheds light on a very important mechanism for neuronal excitability.

 Moreover, some clarification should be done.

  • For instance the authors should better clarify the effect of CKII inhibitor TBCA on mitochondrial influx in hyppocampal neurons stimulated at 2 Hz for 1.3 s to generate an action potential. They call into question the involvement of mitochondrial MCU, but do not give direct evidence of this involvement.

Thank you for this important remark we now revise the MS and relate to it  the effect of the inhibitor on MCU is page 4 of the revised MS.

2)   Another question concerns the effect of CKII blockade on global [Ca2+]i. Possibly some experiment should be done to demonstrate the specificity of TBCA on NCLX activity and [Ca2+]m.

In this respect, in the paragraph “Live fluorescence imaging” the use of Fura2 is reported but any experiment was shown.

Thank you for this suggestion. We have now included a global Ca2+ analysis using fluo4 shown in supplementary figure 1 and page 4 of the results. 

Also for culture cells, experiments in HEK293 should be added, as reported in the “cell culture” paragraph.

In this study, we use SH-SY5Y and neurons that have a developed mitochondrial network and are of physiological relevance

Reviewer 2 Report

The manuscript by Katoshevski at al, " CKII control of axonal plasticity is mediated by mitochondrial Ca2+ via mitochondrial NCLX", presents a novel set of findings through a series of careful and thoughtful investigations. This study was incited by their prior proteomic analysis (Kostic et al., 2015 ) and reveals for the first time that Casein Kinase 2 (CKII) regulates the mitochondrial Na+/Ca2+ exchanger, NCLX. This newly identified signaling cascade has potentially multiple physiological and metabolic implication, some of which are exemplified here and lay an important molecular framework for many future discoveries.   

         The investigation was motivated by their prior identification of a new putative Casein Kinase 2 (CKII) site, the serine 271 of NCLX (Kostic et al., 2015). Here, NCLX mediated mitochondrial Ca2+ efflux is shown to be attenuated by specific blockers of CKII as well genetic preventions of phosphorylation at the serine 271 of NCLX (S271A). Furthermore, the specific CKII blockers are shown to lose their effect on NCLX when only serine 271 of NCLX is mutated, demonstrating the specificity of the approach, and directly supporting the main hypothesis. 

         The work then provides an evolution of physiological and metabolic implications of the newly discovered signaling cascade. First, the work links the main driving force of ATP synthase (Δψ), NCLX, and the regulation of NCLX by CKII.  Second, the work revels new roles of CKII and NCLX in the control of intrinsic excitability properties of hippocampal neurons and their plasticity. 

Overall, the data is very convincing and provide an important new view which will likely be of great interest to the broad readership of Cells. However, there is an imperative need to improve the text and the data presentation. It is difficult to follow the story line as it unfolds. In addition, too often jargon is used instead of plain language that will be much more palatable to the readership.  Suggested text changes are listed below followed by suggested improvements to the data presentation. 

Specific comments:

1) The term "rate-limiting step in [Ca2+]m transport" is presented as a general role but it is highly dependent on the physiological context and thus this generalization is more confusing then helpful. The manuscript would be improved if this term is replaced with a more mechanistic language that would help the readers better understand the context. 

2) Allosteric regulation of NCLX by mitochondrial membrane potential (ΔΨm) is mentioned but not explained. What makes it Allosteric and not thermodynamic should be explained. This term does not have deep enough penetrance into the literature to trivialize the need for explanation.

3)  What is known about CKII is minimally discussed. This seems important given its central role in this investigation. Particularly, how CKII is linked to mitochondrial metabolism.    

4) The discussion section could be significantly improved by subdivisions and subheadings. Now it discusses multiple critical points but is heard to navigate.    

General stylistic comments on figures. 

While all the necessary information is included in the manuscript, it is challenging to understand the main points of figures and how the story unfolds because key information is someplace else in the manuscript. This happens too often. This could be fixed by providing the needed information by annotating the data and the diagrams.  Listed below are revisions that could help (see Specific stylistic comments on figures) .  

The figure captions include very lengthy accounts of the statistical outputs. This makes reading the captions very difficult. A more compact presentation could help as well as moving much of that information into a supplementary information table. 

Specific stylistic comments on figures:

Figure 1A: add caption within the panel to indicate that the protein is NCLX

Figure 1B: add caption within the panel to indicate that the protein is NCLX. Clarify which protein is expected to dock and which to phosphorylate. (Presumably CKII).

Figure 1C: clarify that "ATP" means extracellular application of 10 mM ATP. 

Figure 1C-I. clarify what is "wt". These are not wt cells or mice but rather wt NCLX. 

Figure 2A: It could be helpful to provide addition annotations within the panel to clarify what is "wt", what is "control" and that S271 to A or to D is within NCLX.  

Figure 1B-H. Use the annotations selected for A to clarify what the groups are. 

Figure 3. Apply annotations used in Figure 2 to figure 3. 

Figure 3A. Following FCCP addition, the TMRM signal reduced to levels that approach 0, the applied analysis should be provided. 

Figure 4A. Indicate within the figure what the red fluorescence shows. The white scale bar is nearly invisible and should be enlarged. Can the brightness of the DIC image be increased ? It is otherwise difficult to see the cells.  

Figure 4B. Please annotate to indicate what is "wild type" and what was knocked out to constitute "Knock-out". 

Apply annotations used in Figure 2 to figure 3. 

Figure 5A. Please annotate to indicate what is "wild type" and what was knocked out to constitute "Knock-out". The white scale bar is nearly invisible and should be enlarged.

Figure 6. It is understandable why MPTP is represented as a system that spans the space that includes the outer mitochondrial membrane (OMM), the intermembrane space (IMS), and the matrix. Indeed, there is a great level of uncertainty about the exact whereabout of MPTP. However, the information is much clearer with regards to MCU and NCLX and therefore should be represented accordingly to avoid confusion.

Author Response

Thank you for the helpful and constructive suggestion that we fully addressed :

Comments and Suggestions for Authors

The manuscript by Katoshevski at al, " CKII control of axonal plasticity is mediated by mitochondrial Ca2+ via mitochondrial NCLX", presents a novel set of findings through a series of careful and thoughtful investigations. This study was incited by their prior proteomic analysis (Kostic et al., 2015 ) and reveals for the first time that Casein Kinase 2 (CKII) regulates the mitochondrial Na+/Ca2+ exchanger, NCLX. This newly identified signaling cascade has potentially multiple physiological and metabolic implication, some of which are exemplified here and lay an important molecular framework for many future discoveries.   

         The investigation was motivated by their prior identification of a new putative Casein Kinase 2 (CKII) site, the serine 271 of NCLX (Kostic et al., 2015). Here, NCLX mediated mitochondrial Ca2+ efflux is shown to be attenuated by specific blockers of CKII as well genetic preventions of phosphorylation at the serine 271 of NCLX (S271A). Furthermore, the specific CKII blockers are shown to lose their effect on NCLX when only serine 271 of NCLX is mutated, demonstrating the specificity of the approach, and directly supporting the main hypothesis. 

         The work then provides an evolution of physiological and metabolic implications of the newly discovered signaling cascade. First, the work links the main driving force of ATP synthase (Δψ), NCLX, and the regulation of NCLX by CKII.  Second, the work revels new roles of CKII and NCLX in the control of intrinsic excitability properties of hippocampal neurons and their plasticity. 

Overall, the data is very convincing and provide an important new view which will likely be of great interest to the broad readership of Cells. However, there is an imperative need to improve the text and the data presentation. It is difficult to follow the story line as it unfolds. In addition, too often jargon is used instead of plain language that will be much more palatable to the readership.  Suggested text changes are listed below followed by suggested improvements to the data presentation. 

Specific comments:

1) The term "rate-limiting step in [Ca2+]m transport" is presented as a general role but it is highly dependent on the physiological context and thus this generalization is more confusing then helpful. The manuscript would be improved if this term is replaced with a more mechanistic language that would help the readers better understand the context. 

2) Allosteric regulation of NCLX by mitochondrial membrane potential (ΔΨm) is mentioned but not explained. What makes it Allosteric and not thermodynamic should be explained. This term does not have deep enough penetrance into the literature to trivialize the need for explanation.

3)  What is known about CKII is minimally discussed. This seems important given its central role in this investigation. Particularly, how CKII is linked to mitochondrial metabolism.    

4) The discussion section could be significantly improved by subdivisions and subheadings. Now it discusses multiple critical points but is heard to navigate.    

General stylistic comments on figures. 

While all the necessary information is included in the manuscript, it is challenging to understand the main points of figures and how the story unfolds because key information is someplace else in the manuscript. This happens too often. This could be fixed by providing the needed information by annotating the data and the diagrams.  Listed below are revisions that could help (see Specific stylistic comments on figures).  

The figure captions include very lengthy accounts of the statistical outputs. This makes reading the captions very difficult. A more compact presentation could help as well as moving much of that information into a supplementary information table. 

Specific stylistic comments on figures:

Figure 1A: add caption within the panel to indicate that the protein is NCLX

Thank you for this important note all the captions were corrected according to the reviewer suggestions

Figure 1B: add caption within the panel to indicate that the protein is NCLX. Clarify which protein is expected to dock and which to phosphorylate. (Presumably CKII).

We have now revised it to clarify the phospho site which is relevant to this study.

Figure 1C: clarify that "ATP" means extracellular application of 10 mM ATP. 

Thanks we have revised this part

Figure 1C-I. clarify what is "wt". These are not wt cells or mice but rather wt NCLX. 

We have revised this according to the reviewer's suggestion.

Figure 2A: It could be helpful to provide addition annotations within the panel to clarify what is "wt", what is "control" and that S271 to A or to D is within NCLX.  

Figure 1B-H. Use the annotations selected for A to clarify what the groups are. 

We have revised the figures to clarify this issue as neurons and neurons +TBCA

Figure 3. Apply annotations used in Figure 2 to figure 3. 

Figure 3A. Following FCCP addition, the TMRM signal reduced to levels that approach 0, the applied analysis should be provided. 

Thanks, we revised the MS according to the reviewer's suggestion 

Figure 4A. Indicate within the figure what the red fluorescence shows. The white scale bar is nearly invisible and should be enlarged. Can the brightness of the DIC image be increased ? It is otherwise difficult to see the cells. 

Thanks the red fluorescence indicate GABAergic neurons it is now indicated in the fig 4  

Figure 4B. Please annotate to indicate what is "wild type" and what was knocked out to constitute "Knock-out". 

Apply annotations used in Figure 2 to figure 3. 

We revised fig 4 B and the annotation of B-D are clearer.  

Figure 5A. Please annotate to indicate what is "wild type" and what was knocked out to constitute "Knock-out". The white scale bar is nearly invisible and should be enlarged.

We have revised the captions and annotations of figure 5 according to the helpful reviewer suggestions

Figure 6. It is understandable why MPTP is represented as a system that spans the space that includes the outer mitochondrial membrane (OMM), the intermembrane space (IMS), and the matrix. Indeed, there is a great level of uncertainty about the exact whereabout of MPTP. However, the information is much clearer with regards to MCU and NCLX and therefore should be represented accordingly to avoid confusion.

Thanks for these important remarks we revised the illustration to show that NCLX and MCU are in the inner while MPTP spans the 2 mitochondrial membranes.

Reviewer 3 Report

In this interesting study, Katoshevski et al. demonstrate that CKII mediated phosphorylation of NCLX plays an important role in regulating mitochondrial Ca2+ dynamics. They further show that CKII modulates NCLX susceptibility to depolarization of mitochondrial membrane potential. Finally, they report that NCLX is critical for maintaining axonal plasticity. Overall, this is a sound study wherein conclusions are supported with the data and appropriate controls are used throughout the study. There are some suggestions to improve the quality of data presentation:

1.     Define “AIS” used in the abstract.

2.     Define “DIV” where it is used for the first time in the manuscript.

3.     Include a table with the details of all the abs used in the study.

4.     Instead of bar graphs with error bars, the authors should present scatter plots through the manuscript.

5.     Define “2AP” used in Figure 1G in text and figure legend. As of now, no details are provided.

6.     In Figure 1C, the basal mitochondrial Calcium levels do not appear to be decreased in the representative trace. Please provide a better representative trace that holistic represents data plotted in Figure 1F.

7.     In Figure 2A, include a loading control for blot and provide molecular weight ladder. This blot should be repeated at least 3 times and quantitative densitometry data should be provided.

8.     In Figure 6, “Ca2+ raise” must be correct to “Ca2+ rise/increase”

Author Response

Thank you for the helpful and constructive suggestion that we fully addressed :

Comments and Suggestions for Authors

In this interesting study, Katoshevski et al. demonstrate that CKII mediated phosphorylation of NCLX plays an important role in regulating mitochondrial Ca2+ dynamics. They further show that CKII modulates NCLX susceptibility to depolarization of mitochondrial membrane potential. Finally, they report that NCLX is critical for maintaining axonal plasticity. Overall, this is a sound study wherein conclusions are supported with the data and appropriate controls are used throughout the study. There are some suggestions to improve the quality of data presentation:

  1. Define “AIS” used in the abstract.

Please note that AIS is defined in the abstract and on page 2 of the introduction

  1. Define “DIV” where it is used for the first time in the manuscript.

Thanks for raising this issue. It is defined in page 6 of the results.

  1. Include a table with the details of all the abs used in the study.

Thank you for raising this issue, we have add a table of all the antibodies that were used in this study in the materials and methods section

  1. Instead of bar graphs with error bars, the authors should present scatter plots through the manuscript.

Thanks for the suggestion we revised all relevant figures to scatter plots.

  1. Define “2AP” used in Figure 1G in text and figure legend. As of now, no details are provided.

Following the reviewer's suggestion we have now defined 2AP in figure legend 1 and in page 4 of the results.

  1. In Figure 1C, the basal mitochondrial Calcium levels do not appear to be decreased in the representative trace. Please provide a better representative trace that holistic represents data plotted in Figure 1F.

Thank you for the very helpful advice. Following the reviewer's advice we now show the arbitrary unit fluorescence that shows clearly the  resting basal  Ca2+  level and is quantified in this manner in figure 1 F  

  1. In Figure 2A, include a loading control for blot and provide molecular weight ladder. This blot should be repeated at least 3 times and quantitative densitometry data should be provided.

Following the reviewer's advice, we include several repetitions of the WB add actin as a loading control and quantify them by densitometry. All shown in revised figure 2 A-B

  1. In Figure 6, “Ca2+raise” must be correct to “Ca2+ rise/increase”

Thanks, corrected